# Wind Power Short-Term Forecasting Hybrid Model Based on CEEMD-SE Method

**Keke Wang [1,2,\*], Dongxiao Niu [1,2], Lijie Sun [1,2], Hao Zhen [1,2], Jian Liu [3], Gejirifu De [1] and Xiaomin Xu [1]**

[1]  School of Economics and Management, North China Electric Power University, Beijing 102206, China; ndx@ncepu.edu.cn (D.N.); sunlijie@ncepu.edu.cn (L.S.); mailofzhenhao@163.com (H.Z.); dove@ncepu.edu.cn (G.D.); xuxiaomin0701@126.com (X.X.)
[2]  Beijing Key Laboratory of New Energy and Low-Carbon Development, Beijing 102206, China
[3]  North China Power Dispatching and Control Centre, Beijing 100053, China; liu_jian04@163.com
[\*]  Correspondence: wk915783810@163.com; Tel.: +86-156-5291-2329

**Abstract:** Accurately predicting wind power is crucial for the large-scale grid-connected of wind power and the increase of wind power absorption proportion. To improve the forecasting accuracy of wind power, a hybrid forecasting model using data preprocessing strategy and improved extreme learning machine with kernel (KELM) is proposed, which mainly includes the following stages. Firstly, the Pearson correlation coefficient is calculated to determine the correlation degree between multiple factors of wind power to reduce data redundancy. Then, the complementary ensemble empirical mode decomposition (CEEMD) method is adopted to decompose the wind power time series to decrease the non-stationarity, the sample entropy (SE) theory is used to classify and reconstruct the subsequences to reduce the complexity of computation. Finally, the KELM optimized by harmony search (HS) algorithm is utilized to forecast each subsequence, and after integration processing, the forecasting results are obtained. The CEEMD-SE-HS-KELM forecasting model constructed in this paper is used in the short-term wind power forecasting of a Chinese wind farm, and the RMSE and MAE are as 2.16 and 0.39 respectively, which is better than EMD-SE-HS-KELM, HS-KELM, KELM and extreme learning machine (ELM) model. According to the experimental results, the hybrid method has higher forecasting accuracy for short-term wind power forecasting.

**Keywords:** wind power forecasting; hybrid forecasting model; complementary ensemble empirical mode decomposition; sample entropy; improved extreme learning machine with kernel

## 1. Introduction

With the increasingly serious problem of energy shortage and environmental pollution, energy saving and emission reduction strategy is the key measure to promote green development [1,2], and renewable energy is increasingly favored by people [3]. As one of the most promising renewable energy sources, wind power has been paid worldwide attention owing to its clean, low-carbon and economic characteristics [4,5]. In recent years, China has vigorously developed renewable energy, and both the installed capacity and the grid-connected capacity of wind power are becoming larger and larger. According to the relevant data released by the National Energy Administration, in 2018, China added 21.27 million kW of wind power grid-connected installed capacity [6], and the cumulative grid-connected installed capacity reached 184 million kW, accounting for 9.7% of the total wind power installed capacity. Although wind power generation technology is gradually matured, due to the intermittence, randomness, fluctuation, and uncertainty of wind energy, the large-scale access of wind farms brings tough challenges to the safe and stable operation of power systems [7,8]. Therefore,

accurate and effective wind power forecasting technology is necessary for the security and stability of the power grid. Firstly, accurately forecasting wind power can provide decision support for grid dispatching plan and power market transaction [9], strengthen the stability and safety of the power system [10]; secondly, it can be helpful to the reduction of systems rotating reserve capacity and power generation cost, and the enhancement of the wind farms' economic benefits and the utilization rate of wind power by users [11]; in addition, with the advancement of power market reform, accurate short-term wind power forecasting can also provide relevant basis for the grid-connected wind power sales under power market conditions [3], promote inter-provincial priority to absorb renewable energy, and reduce the risk brought by wind power uncertainty for power market participants [12]. Thus, it follows that wind power forecasting is of great significance to the safe operation of power system and the sound development of the power industry.

At present, scholars all over the world have conducted extensive research on the models and methods of wind power forecasting, mainly including three categories: physical models, statistical models, and intelligent forecasting models [13–15]. The physical model is more complex and relies on meteorological and wind speed data provided by numerical weather prediction (NWP) system, which requires high accuracy and completeness of data, and is more geared for long-term prediction of wind power [16,17]. Although the physical model does not require the support of historical data, NWP data needs a lot of calculations in order to forecast wind power, resulting in fewer engineering applications in China. The statistical methods mainly include regression analysis, exponential smoothing, time series, and Kalman filtering, which are superior to physical methods when forecasting short-term wind power [18]. Statistical models are more mature in the application of wind power forecasting than physical methods by studying the information related to time and space in the data. However, due to the fact that the linear models are commonly used in statistical methods, the randomness of meteorological data cannot be accurately expressed, and the forecasting results of wind power series are poor because of the non-linear and non-stationary characteristics of the series [19,20]. In recent years, artificial intelligence has developed rapidly, and a variety of intelligent forecasting models and machine learning algorithms have been successfully used to wind power forecasting, and good results have been obtained. Common intelligent forecasting models include artificial neural network (ANN) [21], support vector machine (SVM) [22,23], least squares support vector machine (LSSVM) [24], etc. The single intelligent algorithm itself has certain defects [25]. It is difficult to ensure high forecasting accuracy, for it cannot accurately grasp the law of wind power variation, resulting in large errors at the forecasting points where wind power fluctuates sharply. Therefore, some hybrid models of intelligent algorithms have been proposed. And it can be seen from the empirical results that the hybrid model has higher forecasting accuracy and better fitting effect. In addition, because the time series of wind power data are non-stationary sequences with strong randomness and volatility, data pre-processing method can effectively improve the wind power forecasting accuracy [26]. Recently, various data preprocessing methods have been used in the forecasting the short-term wind power to achieve the purpose of feature dimension reduction and data noise reduction. In general, the results are satisfactory. There are some common data preprocessing methods used by researchers, including principal component analysis (PCA) [27], correlation analysis [28], cluster analysis [29], wavelet transform (WT) [30], empirical mode decomposition (EMD) [31] and other corresponding improved methods. For example, the empirical wavelet transform (EWT) is an innovative adaptive wavelet decomposition method. The core idea of EWT is to extract modal signal components with Fourier compact support by constructing orthogonal wavelet filter banks adaptively. Compared with EMD method, it can avoid the phenomenon of mode aliasing and false modes, and get fewer modal signal components.

In wind power forecasting, the common hybrid models usually combine data preprocessing methods with intelligent forecasting models. Through the data preprocessing method, data screening and dimension reduction can be performed to reduce data redundancy. Then, through the signal decomposition algorithm, multiple stable components' sets of wind speed or wind power can develop. After obtaining the forecasting results of each component with intelligent forecasting model, the final

forecasting outcome of wind speed or wind power can be achieved by reconstructing the results. Du et al. [32] used complementary ensemble empirical mode decomposition (CEEMD) to decompose wind power series, by doing so, the noise data can be eliminated, and the main features of original data can be extracted. Then, the improved wavelet neural network (WAN) was used to forecast wind power. Jiang et al. [33] considered that the existing wind speed forecasting models neglect the fluctuation influence between adjacent wind turbines, resulting in poor forecasting accuracy. For this purpose, they constructed a combined wind speed forecasting model, which uses gray correlation analysis to screen the fluctuation information and inputs the obtained fluctuation information into the SVM for wind speed forecasting. Yan et al. [34] proposed a new hybrid wind speed prediction method, which uses correlation-assisted discrete wavelet transform (DWT) for decomposition the wind speed sequence and utilizes generalized autoregressive conditional heteroscedastic (GARCH) to reflect the fluctuation of the subsequence and adjust the parameters of the prediction model based on LSSVM in real time, so as to reflect the change of wind speed better. This method has achieved good results in accuracy and stability. Fei [35] proposed an integrated model for wind speed forecasting using EMD and multi-kernel relevance vector regression (MkRVR) algorithm. The wind speed was decomposed into intrinsic mode functions (IMFs) with different frequency ranges by EMD method, and the forecasting model was established by using MkRVR for the forecasting of each decomposed signal. The empirical results proved that the EMD-MkRVR model is accurate and efficient in predicting wind speed. Khosravi et al. [36] studied the prediction effect of three machine learning algorithm models for wind power output, including multilayer feed-forward neural network (MLFFNN), support vector regression with a radial basis function (SVR-RBF), and adaptive neuro-fuzzy inference system (ANFIS). The input variables of the three models were temperature, pressure, relative humidity and local time. The comparison between the empirical results showed that SVR-RBF was superior to MLFFNN and ANFIS-PSO. Wu and Lin [37] firstly used variational mode decomposition (VMD) to decompose the original wind speed sequence into subsequences of various frequencies. Then bat algorithm (BA) was used to the parameters optimization of least squares support vector machine (LSSVM), and each subsequence was forecast by the improved LSSVM model. Finally, the results of wind speed forecasting were achieved by superimposing the forecasting results of each subsequence. Jiang et al. [38] developed a hybrid wind speed prediction approach based on EMD, VMD and sample entropy (SE). The performance of the hybrid approach was verified by the real measured wind speed data of two cases, which proved that it had better prediction effect. Tian et al. [39] constructed a hybrid wind speed prediction model consisting of data preprocessing, optimization model and prediction module. This model balanced the contradiction between prediction accuracy and stability. In the empirical study, the mean absolute percentage error (MAPE) of the model was less than 4%, which confirmed that it has high prediction accuracy. Yin et al. [40] constructed a hybrid model for short-term wind power forecasting. The extreme learning machine (ELM) optimized by the crisscross optimization algorithm (CSO) was used to forecast, which solved the problem of premature convergence of ELM and had high forecasting accuracy. Through the analysis of the existing literature, it is found that the existing literature pays insufficient attention to data preprocessing. The effective data preprocessing method can improve the input data quality of the model and improve the prediction accuracy. However, VMD, EEMD, CEEMD and other methods will get more subsequences when decomposing the original power series, which increases the complexity of forecasting. As an improved neural network, KELM has the advantage of fast training speed and strong generalization ability. The KELM method optimized by HS has better prediction performance and stronger global search capability.

On account of the above analysis, for the purpose of obtaining more accurate wind power forecasting results, this paper proposes a multi-step hybrid wind power forecasting model combining complete ensemble empirical mode decomposition (CEEMD), sample entropy and improved extreme learning machine with kernel (KELM). The hybrid forecasting models is based on the combination of mixed signal decomposition algorithm and optimized machine learning algorithm, which includes data preprocessing stage, optimizing stage and forecasting stage. Firstly, Pearson correlation analysis

is adopted to screen out the input data which have high correlation with wind power, and extract the main features of the original data, so as to reduce data redundancy. Then, CEEMD method is utilized to decompose the wind power time series into a series of modal components, so as to eliminate noise and improve the quality of input data of the forecasting model. The sample entropy is used to classify and reconstruct subsequences, which can effectively reduce the amount of calculation. Secondly, the harmony search (HS) algorithm is used to optimize the parameters and improve the search ability of KELM model [41]. Finally, the improved KELM is applied to forecast each component, and the forecasting results of respective component are reconstructed to obtain the wind power forecasting value. The KELM model is faster in training and overcomes the shortcomings of easily falling into local optimal solution. In this paper, different Chinese wind farms are used as practical cases, and the multi-step hybrid wind power forecasting model is used to forecast wind power in ultra-short term and short term. Under the same conditions, the EMD-SE-HS-KELM, HS-KELM, KELM and ELM methods are compared to validate the effectiveness of the wind power forecasting model constructed in this paper. The innovations of this paper are as follows:

(1) Ensure the quality of input data of the forecasting model. Most wind power forecasting models only adopt a single data preprocessing method. Thus, the accuracy is limited due to the fact that the complexity of original data is not properly dealt with. This paper first analyses the correlation of indicators, eliminates the indicators with low correlation degree to reduce data redundancy, and then uses CEEMD to decompose wind power to improve the quality of input data of the forecasting model, and use sample entropy (SE), which was proposed by Richman [42] to classify and reconstruct subsequences to reduce the computational complexity.

(2) Realize multi-step forecasting of wind power. The hybrid wind power forecasting model constructed in this paper comprises three parts: data preprocessing, optimization and forecasting. Reasonable multi-level forecasting model can be more supportive for decision-making.

(3) Balance model forecasting accuracy and stationarity. This paper uses a variety of methods to enhance the forecasting accuracy of wind power and improve the stability of the model. The KELM model with faster training speed is used, and the parameters of the KELM model are optimized by the HS algorithm to improve the search performance.

(4) Verify the performance of the forecasting model comprehensively. According to the real measured data of wind farms in China, the ultra-short-term and short-term forecasting of wind power are carried out by adopting the model respectively, and the comprehensive performance of the forecasting model is investigated by calculating four forecasting accuracy evaluation indicators, which confirms the feasibility of multi-scenario application of the model.

The rest of the paper is organized as follows: Section 2 introduces the method used in the wind power forecasting model constructed in this paper. Section 3 establishes a hybrid wind power forecasting model. Firstly, the Pearson correlation coefficient between the original data set and the wind power is calculated, and the data with low correlation degree is eliminated. CEEMD- SE is used to decompose and reconstruct the wind power time series, which can eliminate noise and reduce the amount of computation; and the KELM model optimized by HS wind power is utilized to forecast wind power. Section 4 validates the effectiveness of the model proposed in this paper by comparing it with EMD-SE-HS-KELM, HS-KELM, KELM, and ELM methods. The conclusion is shown in Section 5.

## 2. Materials and Methods

### 2.1. Pearson Correlation Coefficient

The Pearson correlation coefficient is calculated as follows:

$$
\begin{aligned}
P_{x,y} &= \frac{\text{cov}(X,Y)}{\sigma_X \sigma_Y} = \frac{E((X-\mu_X)(Y-\mu_Y))}{\sigma_X \sigma_Y} \\
&= \frac{E(XY)-E(X)E(Y)}{\sqrt{E(X^2)-E^2(X)}\sqrt{E(Y^2)-E^2(Y)}}
\end{aligned}
\tag{1}
$$

where $(X, Y)$ are two different variables. The value of $P_{x,y}$ whose range is $[-1, 1]$ reflects the degree of linear correlation between $Y$ and $X$. The following conclusions can be drawn:

If $|P_{x,y}| \to 1$, it means that the correlation degree between $Y$ and $X$ is stronger; In contrast, if $|P_{x,y}| \to 0$, the correlation degree is weaker, or they are non-linear correlation, or even irrelevant.

## 2.2. Complementary Ensemble Empirical Mode decomposition (EMD)

EMD is widely applied into signal analysis, which was initially proposed by Hibert-Huang as an adaptive data mining methodology [42]. The complex signal is disintegrated into a finite number of intrinsic mode functions (IMFs) components and a residual component through EMD method. The EMD method can be theoretically employed to the decomposition of signals of any time type, as well as signals according to the temporal feature scale of the data itself. It is superior to other methods in handling non-stationary and nonlinear data [42]. However, it also is liable to produce serious defects of mode mixing problem [43], which will affect the accuracy of intrinsic mode functions. CEEMD is an improved algorithm of EMD and it can effectively reduce the occurrence of this phenomenon by using noise characteristics. CEEMD decomposition is based on EMD, and its implementation process is as follows:

(1) Add $N$ pairs of white noise to original sequence to obtain a set containing $2N$ signals. The auxiliary noise is Gaussian white noise $n_i(t)(i = 1, 2, \cdots, N)$ with a mean of 0 and a constant amplitude coefficient $k$. When $N$ ranges from 100 to 300, the value of $k$ is 0.001~0.5 times of the signal standard deviation.

$$\begin{bmatrix} x_{i1}(t) \\ x_{i2}(t) \end{bmatrix} = \begin{bmatrix} 1 & 1 \\ 1 & -1 \end{bmatrix} \begin{bmatrix} x(t) \\ n_i(t) \end{bmatrix} \tag{2}$$

where $x(t)$ is original signal; $n_i(t)$ is auxiliary white noise; $x_{i1}(t)$ and $x_{i2}(t)$ are the signal pairs after the noise is added.

(2) EMD decomposition is performed on the obtained $2N$ signals, and a set of *IMF* components is obtained for each signal. The *j*-th *IMF* component of the *i*-th signal is $IMF_{ij}$, and the residual component is recorded as the last *IMF* component.

(3) The corresponding *IMF* components are averaged to obtain the components of each phase after the original sequence $x(t)$ is decomposed by CEEMD:

$$IMF_j = \frac{1}{2N} \sum_{i=1}^{2N} IMF_{ij} \tag{3}$$

where $IMF_j$ denotes the *j*-th *IMF* obtained through the decomposing process of the original signal by CEEMD.

## 2.3. Sample Entropy Theory

After decomposing the data sequence by CEEMD, $n$ subsequences can be obtained. If the forecasting model is used directly to model and predict each subsequence, the calculation scale will be large. Therefore, the sample entropy theory can be used to analyze the complexity of each subsequence, and the subsequences can be classified and reconstructed, which can lead to the significant reduction of the computational complexity. The SE is initially proposed by Richman [44]. SE is used to measure the possibility of generating new patterns in signals to assess the complexity of time series. A high sequence self-similarity indicates a low sample entropy value; while a complex sample sequence indicates a large sample entropy value.

For the time series $\{x(n)\} = x(1), x(2), \cdots, x(N)$, the calculation consists following steps:

(1) Construct a sequence of vectors $X_m(1), \cdots, X_m(N - m + 1)$ with dimension $m$, where:

$$X_m(i) = \{x(i), x(i+1), \cdots, x(i+m-1)\}, 1 \le i \le N - m + 1 \tag{4}$$

$X_m(i)$ is a vector, composed of $m$ consecutive $x$ values starting from the $i$th point.

(2) The distance $d[X_m(i), X_m(j)]$ between vectors $X_m(i)$ and $X_m(j)$ is defined as the maximum distance between the corresponding elements:

$$d[X_m(i), X_m(j)] = \max_{k=0,\cdots,m-1}\left(\left|x(i+k) - x(j+k)\right|\right) \tag{5}$$

(3) For a specific $X_m(i)$, denote the number of distances between $X_m(i)$ and $X_m(j)$ $(1 \le j \le N - m, j \ne i)$ that are less than or equal to $r$ as $B_i$. For $1 \le j \le N - m$, define:

$$B_i^m(r) = \frac{1}{N - m - 1} B_i \tag{6}$$

(4) Define $B^{(m)}(r)$:

$$B^{(m)}(r) = \frac{1}{N - m} \sum_{i=1}^{N-m} B_i^m(r) \tag{7}$$

$B^{(m)}(r)$ is the probability that two sequences match $m$ points under similar tolerance $r$.

(5) Expand the dimension to $m + 1$. Similar to $B_i$, repeat step (3) and denote the number of conditional distances as $A_i$. Define $A_i^m(r)$:

$$A_i^m(r) = \frac{1}{N - m - 1} A_i \tag{8}$$

(6) Define $A^m(r)$:

$$A^m(r) = \frac{1}{N - m} \sum_{i=1}^{N-m} A_i^m(r) \tag{9}$$

$A^m(r)$ is the probability that two sequences match $m + 1$ points.

The Sample Entropy is defined as:

$$SamEn(m, r) = \lim_{N \to \infty}\left\{-\ln\left[\frac{A^m(r)}{B^m(r)}\right]\right\} \tag{10}$$

When $N$ is a finite value, it can be estimated as follows:

$$SamEn(m, r, N) = -\ln\left[\frac{A^m(r)}{B^m(r)}\right] \tag{11}$$

## 2.4. Harmony Search (HS) Algorithm

HS algorithm is a new optimization algorithm enlightened by musical performance process in bands [41]. It is featured by strong global optimization ability, simple structure and few parameters. The HS algorithm is also popular in the optimization of neural networks. All in all, the procedure of the HS algorithm include initialization, a new harmony generation, update of harmony memory (HM), and judgment of algorithm termination conditions. The calculation of HS algorithm is described as follows:

(1) Initialization

Setting parameters. Parameters comprise harmony memory size (HMS), harmony memory considering rate (HMCR), pitch adjusting rate (PAR), and bandwidth (BW). The meanings of parameters are explained as follows:

HMS: the music played by each instrument has a certain range. A solution space is constructed by the music playing range of each instrument, and then a harmony memory is randomly generated by this solution space.

HMCR: it is necessary to take a group of harmony from this harmony memory through a certain probability, and fine tune this group of harmony to get a group of new harmony, and then judge whether this group of new harmony is better than the worst harmony in the harmony memory, and a memory matching rate needs to be randomly generated.

PAR: select a group of harmonies in the harmony memory with a certain probability to fine tune.

BW: a group of harmonies taken from the memory are tuned with a certain probability, which is specified here.

$X_i = (x_{i1}, x_{i2}, \cdots, x_{id}, \cdots x_{iD})$ represents a harmony vector with D-dimension. The $i$th vector of HM is denoted as $X_i$, and each dimension is generated by the following formula:

$$X_{i,d} = x_{\min,d} + (x_{\max,d} - x_{\min,d}) * rand() \tag{12}$$

where: $d \in [1, D]$ and $i \in [1, HMS]$. $rand()$ represents a random number, which is uniformly distributed in $[0, 1]$. $x_{\min,d}$ and $x_{\max,d}$ represent the maximum value and minimum value of the search range for each dimension variable respectively.

(2) New harmony Generation

Generate a new harmony vector $X_{new}$, and each dimension of it is created as follows:

$$X_{new,d} = \begin{cases} X_{i,d} & if \ r_1 < HCMR \\ X_{new,d} & otherwise \end{cases} \tag{13}$$

$$X_{new,d} = \begin{cases} X_{i,d} \pm rand() * bw_d & if \ r_2 < PAR \\ X_{\min,d} + (X_{\max,d} - X_{\min,d}) * rand() & otherwise \end{cases} \tag{14}$$

where: $X_{i,d}$ is randomly chosen from HM; $bw_d$ is the d-dimensional of $bw$; $r_1$ and $r_2$ are random numbers uniformly distributed in the range $[0, 1]$.

(3) Update of HM

The objective function is used to evaluate the new harmony. Compare the new harmony $X_{new}$ with the worst harmony in the HM, if the former is better, then replace the worst harmony in the HM with the new harmony. For the KELM model, the kernel parameter $\delta$ and the penalty coefficient $C$ are evaluated.

(4) Judgment of algorithm termination conditions

To determine whether the algorithm satisfies the termination condition, the algorithm outputs the best harmony vector, otherwise it returns to step 2.

## 2.5. Extreme Learning Machine with Kernel

Traditional learning algorithms, such as back propagation neural network (BPNN) algorithm, have some shortcomings, like slow training speed and easy to fall into local minimum points. Based on the single hidden layer feed-forward neural network (SLFN), the extreme learning machine (ELM) is a new algorithm, which has been widely used in many fields for its good learning ability. The connection weight between the input layer and the hidden layer is randomly generated as well as the threshold of the hidden layer neurons. The algorithm simply set the hidden layer neuron number in training, and a unique global optimal solution can be attained with quick learning speed and excellent generalization capability. The network structure of the extreme learning machine is shown in Figure 1.

In Figure 1, $\omega_{ij}$ is the connection weight involving $x_i (i = 1, 2, \cdots, n)$ and $v_j (j = 1, 2, \cdots, l)$, while $v_l$ is randomly generated as the threshold of the hidden layer; $\beta_{ij}$ represents the connection weight involving $v_i (i = 1, 2, \cdots l)$ and $y_j (j = 1, 2, \cdots, m)$. For a single hidden layer neural network with a $(n - l - m)$ structure, given $Q$ samples $(x_i, t_i)$, the input data is $x_i = [x_{i1}, x_{i2}, \cdots, x_{im}]^T$, the expected output is $t_i = [t_{i1}, t_{i2}, \cdots, t_{im}]^T$. The output of the network ELM is [45]:

$$y_j = \sum_{i=1}^{l} \beta_i \sigma_i (\omega_i x_j + v_j), j = 1, 2, \cdots, Q \tag{15}$$

where: $\sigma_i$ is the activation function.

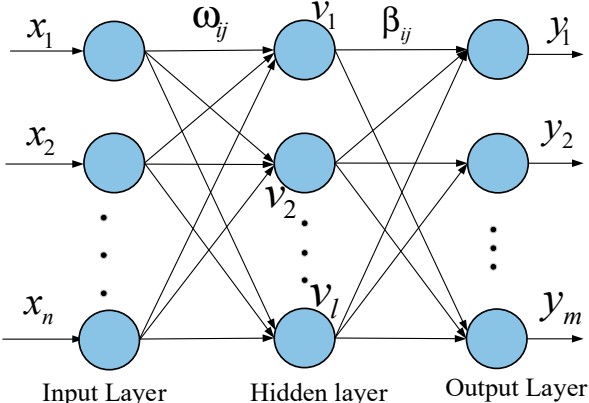

**Figure 1.** Extreme learning machine (ELM) training model with $(n - l - m)$ structure.

$$H_{\omega,v,x} \beta = T, \beta = \begin{bmatrix} \beta_1^T \\ \vdots \\ \beta_l^T \end{bmatrix}, T = \begin{bmatrix} t_1^T \\ \vdots \\ t_m^T \end{bmatrix} \tag{16}$$

where: $T$ is the expected output vector; $H_{\omega,v,x}$ is denoted as the output matrix of the hidden layer, which can be expressed as:

$$H_{\omega,v,x} = \begin{pmatrix} \sigma(\omega_1 x_1 + v_1) & \cdots & \sigma(\omega_l x_1 + v_l) \\ \vdots & \ddots & \vdots \\ \sigma(\omega_1 x_Q + v_1) & \cdots & \sigma(\omega_l x_Q + v_l) \end{pmatrix} \tag{17}$$

Calculate the solution by means of the Moore-Penrose generalized inverse [46]:

$$\beta^* = H^+ T \tag{18}$$

On the basis of ELM, Huang combines nuclear learning with ELM, replaces random mapping in ELM with nuclear mapping, and proposes KELM algorithm [47]. By introducing the kernel function to obtain better application potential, the definition of kernel function is as follows:

$X$ is the input space, $H$ is the feature space, if there is a mapping $\varphi(x) : X \to H$ from $X$ to $H$, so that for all $x, y \in X$, function $K(x, y) = \varphi(x) \cdot \varphi(y)$, then $K(x, y)$ is the kernel function, $\varphi(x)$ is the mapping function, and $\varphi(x) \cdot \varphi(y)$ is the inner product of $x, y$ mapped to the feature space [48]. This method achieves better application by introducing a kernel function. Defines the kernel matrix as:

$$\begin{cases} Q_{ELM} = HH^T \\ Q_{ELM_{ij}} = h(x_i)h(x_j) = K(x_i, x_j) \end{cases} \tag{19}$$

where: $h(x)$ denotes the output function of the hidden layer node; $K(x_i, x_j)$ is a kernel function, namely:

$$K(x_i, x_j) = \exp\{-\|x_i - x_j\|/2\delta^2\} \tag{20}$$

According to Equations (19) and (20) he output and output weights of KELM are as follows:

$$f(x) = \begin{bmatrix} K(x, x_1) \\ \vdots \\ M(x, x_Q) \end{bmatrix} \left( \frac{I}{C} + \Omega_{ELM} \right)^{-1} y \tag{21}$$

$$\beta = \left( \frac{I}{C} + \Omega_{ELM} \right)^{-1} y \tag{22}$$

This paper utilize the HS algorithm to optimize the kernel parameter δ and the penalty coefficient C, which is crucial for the search ability of the KELM algorithm.

Extreme Learning Machine with Kernel (KELM) has strong generalization ability with a high learning speed. Combined with the SLFN of the kernel learning map, this method overcomes the shortcomings of the traditional neural network, which easily falls into the local optimal solution.

## 3. Wind Power Forecasting Model

### 3.1. Model Design

This section introduces the development of wind power forecasting model, including model design and model comprehensive evaluation. The flow chart of the CEEMD-SE-HS-KELM model is presented in Figure 2. The hybrid model is composed by three parts: data preprocessing, optimization and prediction phase.

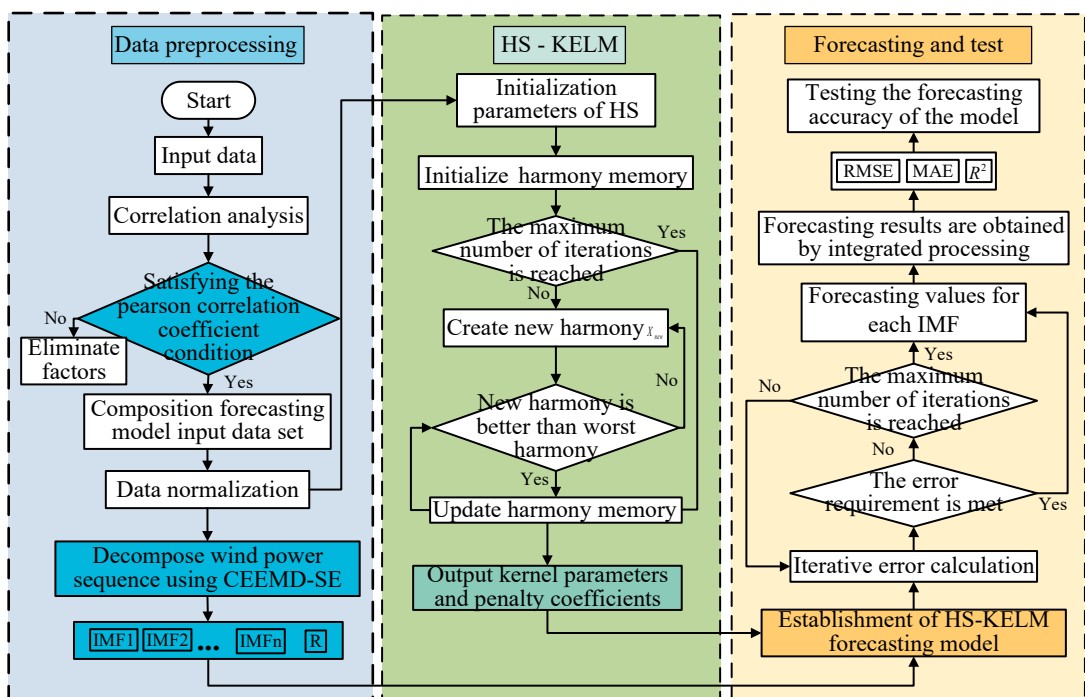

**Figure 2.** The flow chart of CEEMD-SE-HS-KELM model.

The time series of wind power has strong non-linearity and non-stationarity. CEEMD is used to decompose wind power, reduce the fluctuation of original data, and overcome the modal mixture phenomenon of EMD to obtain multiple IMF components. Then, the complexity of each subsequence is analyzed by SE, and the reconstruction of the subsequence is carried out to reduce the amount of calculation, and HS-KELM model is utilized to forecast the IMF components. Eventually, the final

wind power forecasting results are obtained through integrated processing. This paper establishes a CEEMD-SE-HS-KELM model to forecast wind power, as shown below:

(1) Screen the original data set through correlation analysis to obtain data indicators with high correlation, which can used as the input data of CEEMD-HS-KELM forecasting model;

(2) Decompose the original wind power sequence *X* through CEEMD method to obtain *n* sub sequences components from high frequency to low frequency, which are $n - 1$ intrinsic mode function $IMF_i(t)$ and one approximately monotonous residual $R(t)$;

(3) Utilize SE theory to calculate the complexity of each subsequence and reconstruct the subsequence decomposed by CEEMD;

(4) HS-KELM model is constructed for each subsequence and the forecast values of each subsequence are obtained;

(5) Superimpose the wind power forecasting results of each subsequence to obtain the final forecasting results.

*3.2. Evaluation Criteria*

For the performance test of the proposed model, three evaluation criteria are of great importance, including Root Mean Square Error (RMSE), Mean Absolute Error (MAE) and Determination Factor R$^2$. The calculation methods are as follows:

$$RMSE = \sqrt{\frac{1}{n}\sum_{i=1}^{n}\left(p_i - p_i'\right)^2} \tag{23}$$

$$MAE = \frac{1}{n}\sum_{i=1}^{N}\left|p_i' - p_i\right| \tag{24}$$

$$R^2 = \frac{\sum\limits_{i=1}^{n}\left(p_i' - \overline{p_i}\right)^2}{\sum\limits_{i=1}^{n}\left(p_i - \overline{p_i}\right)^2} \tag{25}$$

where: $p_i$ represents the true power value, kW; $p_i'$ represents the anti-normalized power value after CEEMD-SE-HS-KELM output, kW; *n* represents the data number.

## 4. Short-Term Wind Power Forecasting

To comprehensively verify the performance of the CEEMD-SE-HS-KELM wind power hybrid forecasting model proposed in this paper, the following case is designed. Taking the measured data of a wind farm in China as an example, short-term wind power is made by the hybrid forecasting model, and compared with many intelligent forecasting models, multiple error indicators are calculated, and the fitting effect and forecasting accuracy of the model are analyzed.

The wind power data of Beijing wind farm in May 2017 is selected as experimental sample. The rated installed capacity of the wind farm is 49 MW and the sampling frequency is one point per 5 min. The data of 17 consecutive days from May 15th to 31st are taken, totaling 4608 sampling points, of which the first 4322 sampling points are training sets of the forecasting model. The last 288 sampling points are the test set, the original data is in the "Supplementary Materials".

*4.1. Data Set Screening*

The original data sets are divided into two categories, one is historical meteorological data, including historical wind power data and historical weather data such as wind speed and wind

direction. According to the Pearson correlation coefficients analysis of the original data sets, the observed correlations degrees between the meteorological data and wind power are shown in Table 1.

**Table 1.** Pearson correlation coefficient analysis results.

| Indicators | Correlation | Indicators | Correlation | Indicators | Correlation |
|---|---|---|---|---|---|
| | 10 m | 0.788 | | 10 m | −0.391 | temperature | 0.227 |
| wind speed | 30 m | 0.796 | wind direction | 30 m | −0.308 | humidity | −0.51 |
| | 50 m | 0.777 | | 50 m | −0.025 | rainfall | 0.29 |
| | 70 m | 0.764 | | 70 m | 0.289 | pressure | −0.37 |
| | hub height | 0.764 | | hub height | 0.289 | - | |

According to the Pearson correlation coefficients, the coupling degree between wind speed, wind direction and wind power are relatively high, so the wind speeds of different heights, the wind direction of 30 m, the wind direction of 50 m are selected as the input variables of the model. The original data is converted into a number between [0, 1], the purpose of which is to eliminate the magnitude difference between the various dimensions of data and avoid the influence on the forecasting result caused by the large magnitude difference between input and output data. The normalization method is as follows:

$$x^*_i = \frac{x_i - x_{\min}}{x_{\max} - x_{\min}} \tag{26}$$

where: $x_{\min}$ and $x_{\max}$ are respectively the minimum and the maximum value in the sequence, $x_i$ is the initial input data, and $x^*_i$ is the normalized data.

### 4.2. CEEMD Decomposition of Wind Power Sequence and Subsequence Reconstruction

The original wind power data is decomposed by CEEMD decomposition method, and 12 IMF components and 1 residual component are obtained. The result of decomposition is shown in Figure 3. After CEEMD decomposition processing, the characteristic changes of the signal are extracted from high frequency to low frequency, and the components are relatively stable.

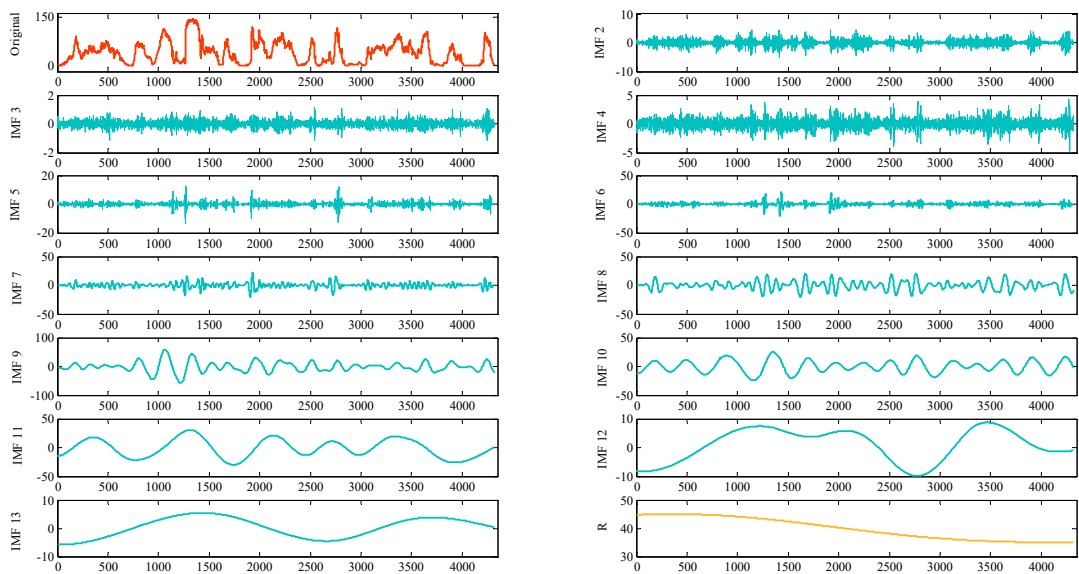

**Figure 3.** Decomposition results of CEEMD.

Using the HS-KELM model to directly forecast each subsequence will increase the amount of computation. SE method is used to calculate the complexity of each subsequence, and the results are shown in Figure 4.

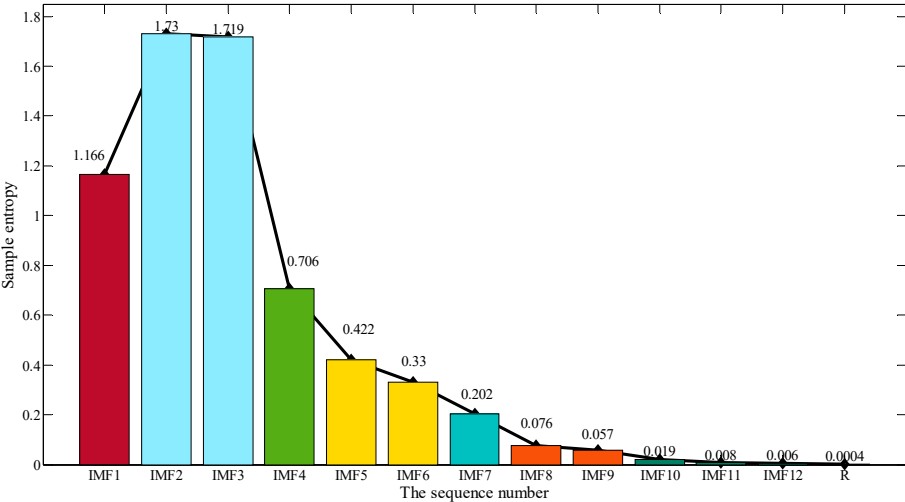

**Figure 4.** Sample entropy of each subsequence.

The criterion of SE for IMF sequence classification is about 0.2 times of the original sequence standard deviation [49]. By calculating the standard deviation of subsequence complexity, the standard deviation is 0.25, so the similarity difference of subsequence reconstruction is obtained. The value is 0.12 and is rounded down to 0.1. As shown in Figure 5, the sample entropy of IMF2 and IMF3 subsequences has small difference, and the two subsequences can be classified into one class, and integrated and reconstructed as a new subsequence input into the HS-KELM for training and forecasting. All subsequences are categorized and the results are presented in Table 2 and the reconstructed results are presented in Figure 5.

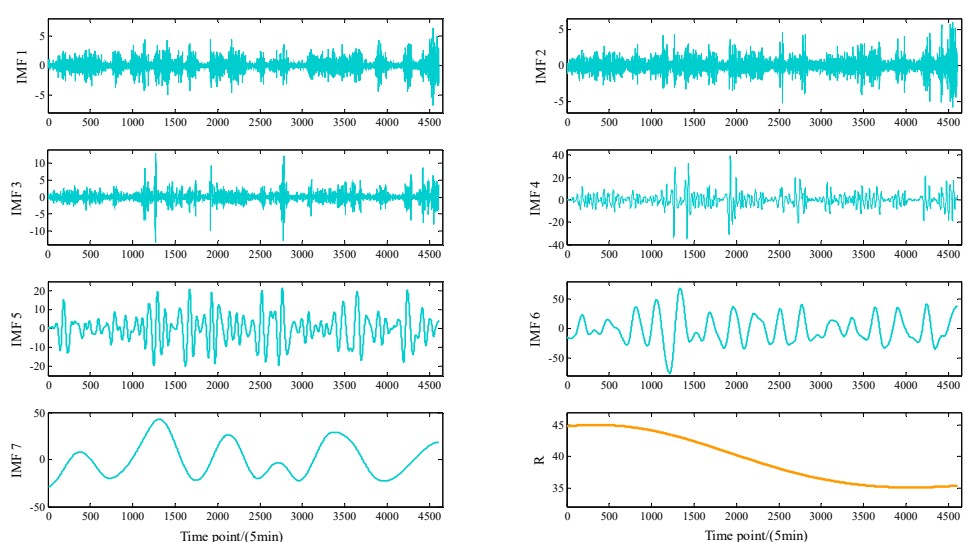

**Figure 5.** Wind power sub-sequences processed by CEEMD-SE.

**Table 2.** Results of the new subsequence with merged intrinsic mode functions (IMF) components.

| New Subsequence Number | Initial Subsequence Number | New Subsequence Number | Initial Subsequence Number |
|:---:|:---:|:---:|:---:|
| 1 | 1 | 5 | 7 |
| 2 | 2,3 | 6 | 8,9 |
| 3 | 4 | 7 | 10,11,12 |
| 4 | 5,6 | 8 | R |

### 4.3. Wind Power Forecasting by CEEMD-SE-HS-KELM Model

Following decomposition and reconstruction for the sequences, 8 HS-KELM forecasting models are constructed, and the above 8 subsequences are respectively trained and forecast. For the HS-KELM model, the common values of 3 parameters *HMCR*, *PAR* and *BW* of HS algorithm is $[0.63, 0.99]$, $[0.01, 0.73]$ and $[0.0004, 0.3]$ respectively [41,50], the maximum iteration limit is detailed in reference [51], and the objective function is set as RMSE. Therefore, $HMS = 100$, $HMCR = 0.9$, $PAR = 0.35$, $BW = 0.25$, the number of new harmony vectors generated each time is 10.

For the HS-KELM model, the number of hidden layer neurons is set to 20, the data of a certain day is randomly selected and the wind power is forecast. RMSE is selected as the objective function, and the number of terminations is set as 100. As shown in Figure 6, the target function value tends to be gentle after 21 iterations, and ends at the 100th generation. The convergence speed is faster, and it is more consistent with the actual power curve. Therefore, the forecasting model structure of HS-KELM model is set as "6-20-1". Train and forecast the eight subsequences respectively, and the final forecasting results of each subsequence are integrated to get the wind power forecasting output value, as shown in Figure 7.

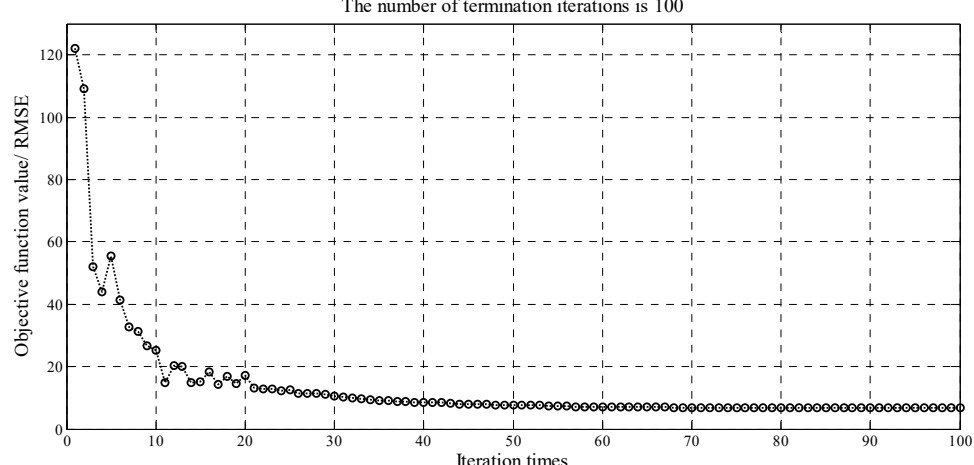

**Figure 6.** Iteration curve.

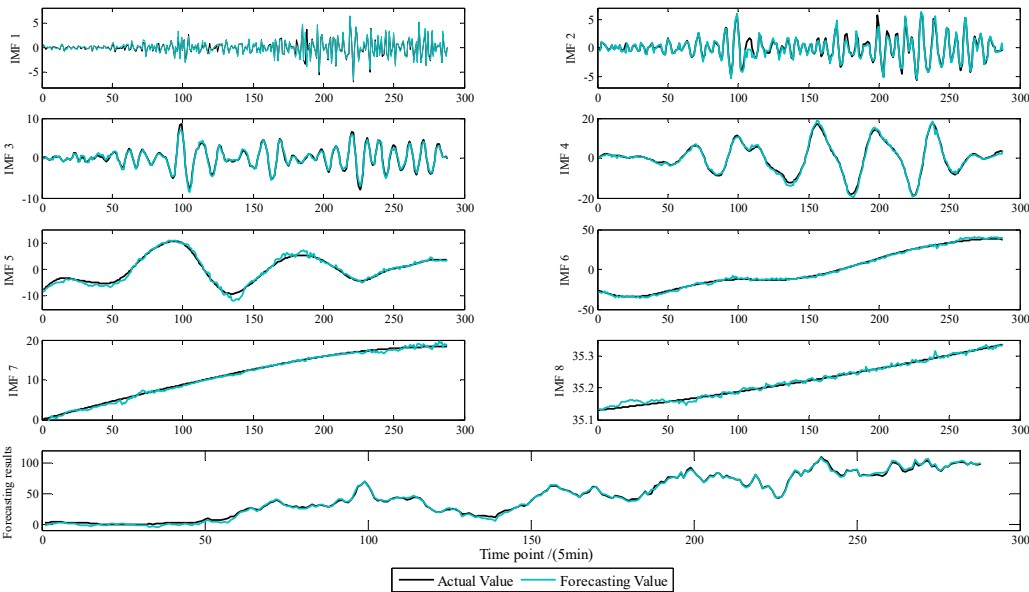

**Figure 7.** Wind power forecasting results.

### 4.4. Comparative Analysis of Forecasting Models

To confirm the validity of this model, ELM, KELM, HS-KELM, EMD-SE-HS-KELM, CEEMD-SE-HS-KELM models are constructed for comparative analysis, and they are respectively named configuration #1–#5. The KELM structure is in the form of "6-20-1", and the maximum number of training time is 500. To avoid the influence of randomness on the forecasting results, the average value of each model is taken after 50 independent runs and compared with the actual wind power. The forecasting results are presented in Figure 8.

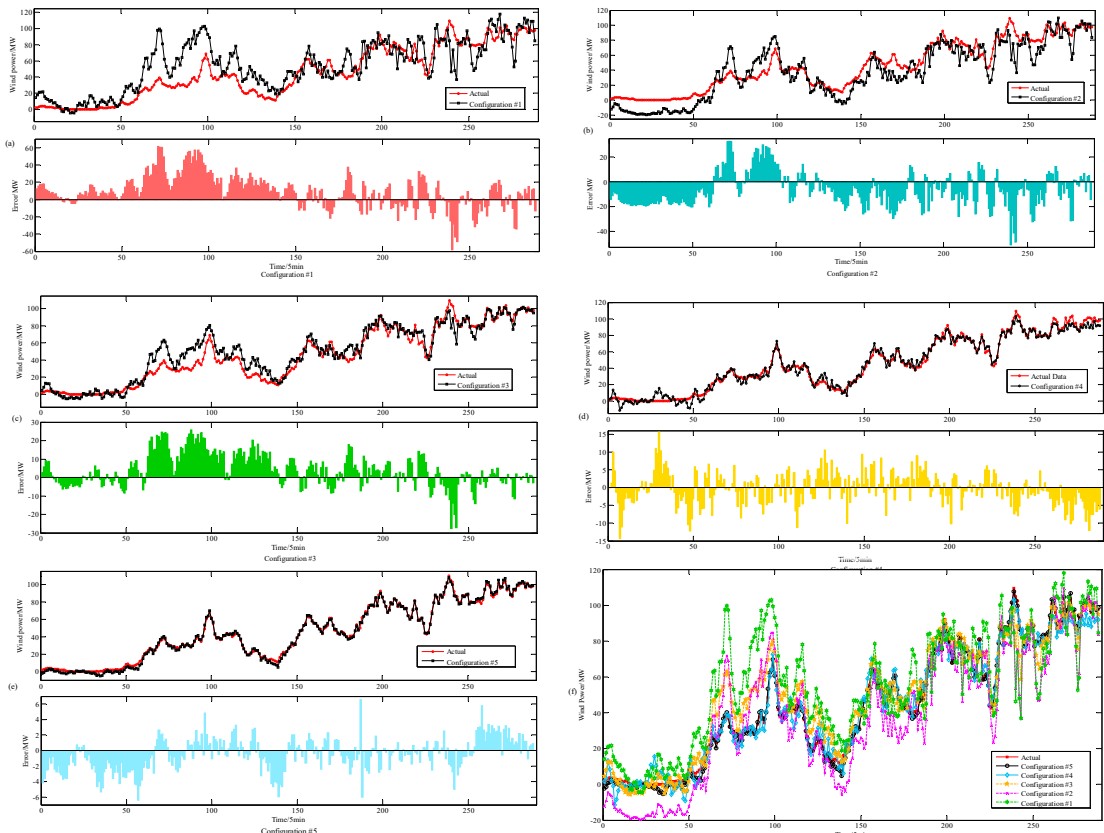

**Figure 8.** Wind power forecasting results (5 min). (**a**) configuration #1: ELM model; (**b**) configuration #2: KELM model; (**c**) configuration #3: HS-KELM model; (**d**) configuration #4: EMD-SE-HS-KELM model; (**e**) configuration #5: CEEMD-SE-HS-KELM model; (**f**) comparison of forecasting values of different models.

To compare and evaluate forecasting accuracy of different models, determination coefficient $R^2$ and model error indexes such as Root Mean Square Error (RMSE) and Mean Absolute Error (MAE) are calculated respectively, as shown in Table 3 and Figure 9.

**Table 3.** The calculation results of model evaluation index.

| Evaluation Index | ELM | KELM | HS-KELM | EMD-SE-HS-KELM | CEEMD-SE-HS-KELM |
|---|---|---|---|---|---|
| **Configuration** | **#1** | **#2** | **#3** | **#4** | **#5** |
| RMSE | 20.84 | 15.63 | 9.74 | 4.67 | 2.16 |
| MAE | 9.29 | 7.05 | 3.86 | 0.52 | 0.39 |
| $R^2$ | 1.12 | 1.05 | 0.96 | 1.03 | 1.01 |

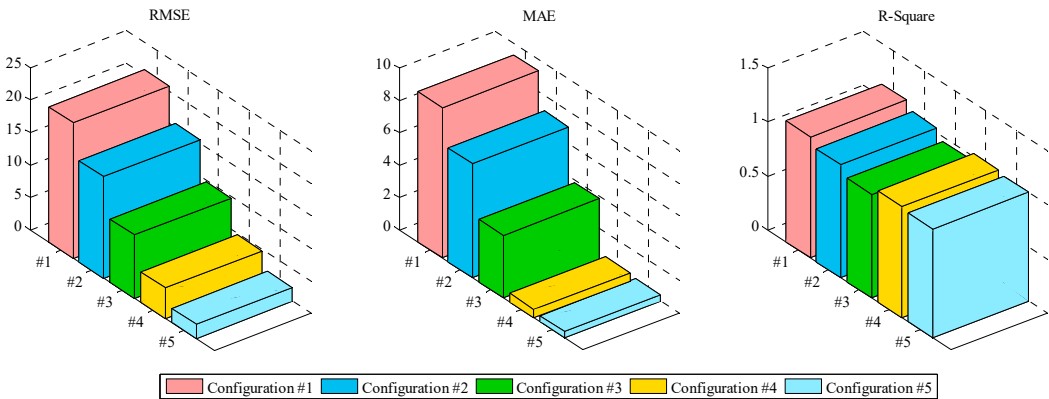

**Figure 9.** Root Mean Square Error (RMSE) and Mean Absolute Error (MAE) and R$^2$ of different forecasting models.

It can be seen from Table 3 and Figure 8, Figure 9:

(1) Comparing EMD-SE-HS-KELM and CEEMD-SE-HS-KELM, the RMSE and MAE of the latter are improved by 53.63% and 24.36% respectively compared with EMD-SE-HS-KELM, which indicates that the hybrid model of data preprocessing based on CEEMD-HS has better processing effect.

(2) Comparing HS-KELM and EMD-SE-HS-KELM, the RMSE and MAE of the latter are improved by 52.10% and 86.58%, respectively, compared with HS-KELM, which indicates that for non-stationary wind power series, pre-processing can effectively eliminate noise, ensure data quality and improve forecasting accuracy.

(3) Comparing KELM and HS-KELM, the RMSE and MAE of the latter are improved by 37.67% and 45.23%, respectively, compared with KELM, which indicates that the parameters of KELM algorithm are optimized by HS algorithm, which effectively improves the search ability and the forecasting accuracy.

(4) Compared with ELM, the RMSE and MAE of the KELM are improved by 25.02% and 24.06% respectively compared with ELM, which indicates that the forecasting accuracy of KELM algorithm is better than ELM model, and KELM has a stronger generalization ability.

It can be seen from Figure 8, that a single KELM and ELM forecasting model can only roughly reflect the wind power trend, but the forecasting value at each time point is quite different from the actual. The CEEMD-SE-HS-KELM hybrid forecasting model constructed in this paper has a good fitting effect between the forecasting result and the actual value at each time point, and its forecasting accuracy is higher.

## 5. Conclusions

Aiming at short-term forecasting of non-linear and unsteady wind power time series, a hybrid forecasting model consisting of CEEMD-SE and HS optimized KELM is proposed in this paper. Firstly, the Pearson correlation coefficient is used to screen the input data to reduce data redundancy. Secondly, the combined data preprocessing strategy of CEEMD-SE is utilized to process the data. The wind power time series is decomposed and reconstructed to eliminate data noise and reduce the computational load. After that, the KELM model optimized by HS is used to forecast each subsequence after reconstructing, and the final wind power forecasting value is obtained after integration processing. Finally, a specific wind farm in China is taken as an example, the CEEMD-SE-HS-KELM, EMD-SE-HS-KELM, HS-KELM, KELM and ELM models are established to forecast wind power respectively. The case study shows that:

(1) KELM model has higher forecasting accuracy than ELM model, and has broad application prospects in wind power forecasting.

(2)   Compared with the single KELM model, HS can optimize the kernel parameters and penalty function of KELM to obtain higher forecasting accuracy, which indicates that HS-KELM model has stronger global search ability and more stable forecasting performance.

(3)   Compared with EMD-SE, the data preprocessing strategy based on CEMD-SE has better performance and effectively improves the forecasting accuracy. The hybrid model proposed in this paper can be well applied to short-term wind power forecasting.

**Supplementary Materials:** The following are available online at http://www.mdpi.com/2227-9717/7/11/843/s1.

**Author Contributions:** All of the authors have contributed to this research. K.W., L.S. and H.Z. collected the data and wrote this paper; D.N. provided professional guidance; J.L. collected the data, G.D. and X.X. revised this manuscript. All authors have approved the submitted manuscript.

**Funding:** This work was supported by the 2018 Key Projects of Philosophy and Social Sciences Research, Ministry of Education, China (grant number 18JZD032); 111 Project, (grant number B18021); Natural Science Foundation of China (grant number 71804045).

**Conflicts of Interest:** The authors declare no conflict of interest.

## Nomenclature

| | |
|---|---|
| NWP | Numerical weather prediction |
| ANN | Artificial neural network |
| SVM | Support vector machine |
| LSSVM | Least squares support vector machine |
| PCA | Principal component analysis |
| WT | Wavelet transform |
| EMD | Empirical mode decomposition |
| EWT | Empirical wavelet transform |
| CEEMD | Complementary ensemble empirical mode decomposition |
| WAS | Wavelet neural network |
| DWT | Discrete wavelet transform |
| GARCH | Generalized autoregressive conditional heteroscedastic |
| MkRVR | Multi-kernel relevance vector regression |
| IMF | Intrinsic mode functions |
| MLFFNN | Multilayer feed-forward neural network |
| SVR | Support vector regression |
| RBF | Radial basis function |
| ANFIS | Adaptive neuro-fuzzy inference system |
| PSO | Particle swarm optimization |
| VMD | Variational mode decomposition |
| BA | Bat algorithm |
| ELM | Extreme learning machine |
| CSO | Crisscross optimization algorithm |
| KELM | Extreme learning machine with kernel |
| HS | Harmony search |
| SE | Sample entropy |
| HM | Harmony memory |
| HMS | Harmony memory size |
| HMCR | Harmony memory considering rate |
| PCR | Pitch adjusting rate |
| BW | Bandwidth |
| BPNN | Back propagation neural network |
| RMSE | Root mean square error |
| MAE | Mean absolute error |
| $R^2$ | Determining factor |

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
