# Peer review of "Wind Power Short-Term Forecasting Hybrid Model Based on CEEMD-SE Method"

_processes, doi:10.3390/pr7110843_

Round 1

Reviewer 1 Report

From my point of view, (i) the introduction provides sufficient background and include all relevant references, (ii) the research design is appropriate, (iii) the methods are adequately described, (iv) the results clearly presented, and (v) the conclusions are supported by the results. Furthermore, Originality / Novelty,  Significance of Content, Quality of Presentation, Scientific Soundness, Interest to the readers and Overall Merit is average.

However, how the paper contributes to bridge the gap of existing knowledge is not clear enough and should be clarified.

Author Response

Dear editor,

Thank you for your letter and for the reviewers’ comments concerning our manuscript. Those comments are all valuable and very helpful for revising and improving our paper, as well as the important guiding significance to our researches. We have studied comments carefully and have made correction which we hope meet with approval. Revised portion are marked in red in the paper. The main revisions are as follows:

(1) We have added the contribution of the paper to the existing research at the end of the introduction, as shown in line 131-137.

(2) We have added some references and modified the inappropriate citation format in the paper.

(3) We have corrected some misspellings and inappropriate sentences in the paper.

(4) We have modified most of the figures, improved the quality of the figures and helped the readers better understand them.

(5) We have added the reason for the selection of some parameters in the paper, and explained and justified our choices.

(6) We have added nomenclature with acronyms and symbols in appendix to avoid confusion.

The responds to the reviewer’s comments are as follows:

We feel deeply grateful for your suggestions, which help us a lot in reconstructing our paper structure to a higher level and revising the manuscript in every detail.

Comment 1: How the paper contributes to bridge the gap of existing knowledge is not clear enough and should be clarified.

Response: We have added the contribution of the paper to the existing research at the end of the introduction, as shown in line 131-137.

Reviewer 2 Report

Dear authors,

I did find the wind forecasting approach suggested in your manuscript very interesting and was agreeably surprised by the results presented. Overall, I think this paper could be of interest to the scientific community and to decision makers in the domain. Nonetheless, the work to bring this paper to a "ready to publish" level is important.

For the reader's sake, numerous modifications should be made to the content of the manuscript. As you know, there is an important quantity of acronyms used through the text which could be really confusing at some point.

-------------

Here is a suggestion that may improve readability :

I'd consider referring to the diverse models in relation to their complexity as From my point of view, each of the compared configuration from the most basic ELM to your configuration (CEEMD-SE-HS-KELM) are mostly built of added layers of complexity. If so, instead of ELM, KELM, HS-KELM, EMD-SE-HS-KELM & CEEMD-SE-HS-KELM, you could refer to them as Configuration #1 through #4 + your model. I think that would render a manuscript easier for the reader to understand as the configuration number would relate to the complexity of the concerned wind forecasting model without the need for confusing acronyms. They would only need to be referred in a table in the methodology section.

-------------

The introduction section is well written and presents the relevant context to your study.

The section that would benefit the most from modifications is the methodology section which needs some clarifications and referencing work to be brought up to ensure that your approach is well understood.

There is also an extensive review of English that needs to be done to ensure the manuscript quality.

You will find my suggestions and comments in the following section :

SPECIFIC COMMENTS

Line 23 : RMSE not RSME. Line 36 : You refer to data from the National Energy Administration, please provide a reference. Line 95 : [...] wavelet neural network WAS used to forecast [...] Line 116 : Then bat algorithm (TYPO). Line 122 : Instead of Reference [36], you should use Tian et al. [36]. Line 141 : This is the first place in the manuscript where you refer to the Harmony Search Algorithm which is a major element in your suggested methodology. You definitely should provide a reference here. Also, add the acronym (HS) to refer to this method further in the text. Line 141 : Remove "of KELM model" as you repeat it further in the same sentence. Line 163 : Refer to the harmony search algorithm as HS. Line 164 : This sentence is incomplete and makes no sense as it is actually written. Lines 180 to 190 : As the Pearson Correlation Coefficient is widely known, I do not think it needs to be presented in such details in the methodology. Line 193 : Add a reference for Hibert-Huang. Line 197 : You state that the EMD method has an advantage for dealing with non-stationarity. I think this assessment should be backed up by literature. Consider adding a reference. Line 198 : Remove the (YEH et al., 2011). Line 222 : [...] The sample entropy (SE) is proposed by Richman [39],  SE measures [...]. The Richman [39] reference should be cited before, see comment on line 163. Line 258 : I think the parameters need to be explained in a more detailed fashion. Line 274 : "better than the worst harmony", based on what criteria ? Line 281 : Define the BP algorithm acronym as it is used for the first time in the manuscript. Line 301 : Moore-Penrose generalized inverse, provide reference. Line 303 : Huang, provide reference. Line 304 : Nuclear mapping, please define and provide reference. Line 308 : Remove "choose". Line 310 : You are not referring to the right equation numbers here. Line 318 : Replace [...] which is easy to fall [...] by [...] which easily falls [...] Line 326 (Figure 2) : This figure should be optimized and take an entire page if necessary as it is a core element of the explanation of your method. Line 339 : replace 1 by one. Line 340 : Replace "Sample entropy" by SE. Line 348 : Determination instead of Determining Line 363 : What year were the data acquired ? And what does -5.31 means ? Line 365 : On a total of 4608 points, you used a subset of 4322 as training points, which corresponds to 92% of the entire dataset. Isn't it a very large proportion to use as training, leaving only 8% on validation ? Line 367 : Where do the historical meteorological data come from ? Add a reference. Line 369 : Pearson, not Person. Lines 369-370 : [...] original data sets, the OBSERVED correlations degrees between the meteorological data and wind power ARE shown in Table 1. [...] Line 372 : Explain in text why the correlation coefficients differ considerably as we change height in link with wind direction. Line 373 : Same as comment on line 369. Line 382 : "CEEMD decomposition of ... " Line 383 : Use "approach" instead of "technology". Line 388 (Figure 3) : Identify your X-axis. (Same for figures 5 and 6) Line 389 : This sentence is a repetition of what was already said in the previous paragraph. Consider removing it. Lines 397-398 : Explain the method used when you proceed to the integration of two neighbor IMF's with a difference smaller than 0.1. Line 405 : "Following decomposition and reconstruction ... ". Lines 406-409 : You list numerous parameters but you don't explain and justify your choices. This is an important detail that needs to be clarified. Lines 416-417 : Same as previous comment. Explain and justify the choices that were made. Line 422 (Figure 7) : Remove the black frames and identify your figures with letters from A to F, like in the figure caption.

Author Response

Dear editor,

Thank you for your letter and for the reviewers’ comments concerning our manuscript. Those comments are all valuable and very helpful for revising and improving our paper, as well as the important guiding significance to our researches. We have studied comments carefully and have made correction which we hope meet with approval. Revised portion are marked in red in the paper. The main revisions are as follows:

(1) We have added the contribution of the paper to the existing research at the end of the introduction, as shown in line 131-137.

(2) We have added some references and modified the inappropriate citation format in the paper.

(3) We have corrected some misspellings and inappropriate sentences in the paper.

(4) We have modified most of the figures, improved the quality of the figures and helped the readers better understand them.

(5) We have added the reason for the selection of some parameters in the paper, and explained and justified our choices.

(6) We have added nomenclature with acronyms and symbols in appendix to avoid confusion.

The responds to the reviewer’s comments are as follows:

Comment 1: As you know, there is an important quantity of acronyms used through the text which could be really confusing at some point.

Response: Thank you very much for your valuable suggestion. We have added nomenclature with acronyms and symbols in appendix to avoid confusion.

Comment 2: Suggestion that may improve readability.

Response: Thanks for your helpful suggestion, which is essential for us to improve the quality of our paper. We have modified ELM, KELM, HS-KELM, EMD-SE-HS-KELM & CEEMD-SE-HS-KELM as Configuration #1 through # 5, and modified Figure 7 and Figure 8 to improve the readability of the paper.

Comment 3: Specific comments: This figure should be optimized and take an entire page if necessary as it is a core element of the explanation of your method.

Response: Thanks for your helpful suggestion, which is essential for us to improve the quality of our paper. We have modified the figure 2, and improved the quality of it.

Comment 4: Specific comments: On a total of 4608 points, you used a subset of 4322 as training points, which corresponds to 92% of the entire dataset. Isn't it a very large proportion to use as training, leaving only 8% on validation?

Response: Thanks for your helpful suggestion, which is essential for us to improve the quality of our paper. When we write, we refer to a lot of papers about wind power forecasting, in which about 10% of the data are often taken for verification. Therefore, we used the wind power data in one day as the verification set to verify the validity of the CEEMD-SE-HS-KELM forecasting model constructed in this paper.

Comment 5: Specific comments: Where do the historical meteorological data come from? Add a reference.

Response: Thanks for your helpful suggestion, which is essential for us to improve the quality of our paper. Historical meteorological data comes from the meteorological statistics of the wind farm, not from the local meteorological bureau, so we are sorry that we cannot provide the reference link of data source here.

Comment 6: Specific comments: Explain in text why the correlation coefficients differ considerably as we change height in link with wind direction.

Response: We’re sorry that due to our mistake, we failed to fill in the content in Table 1 accurately. The value of correlation coefficient has been modified. The correlation between wind direction and wind power is weak. The input data of forecasting model includes wind speed at different heights. Ask for your understanding again for our negligence.

Comment 7: Specific comments: Explain the method used when you proceed to the integration of two neighbor IMF's with a difference smaller than 0.1.

Response: Thanks for your helpful suggestion, which is essential for us to improve the quality of our paper. The criterion of SE for IMF sequence classification is about 0.2 times of the original sequence standard deviation [49]. By calculating the standard deviation of subsequence complexity, the standard deviation is 0.25, so the similarity difference of subsequence reconstruction is obtained. The value is 0.12 and is rounded down to 0.1.

Comment 8: Specific comments: You list numerous parameters but you don't explain and justify your choices. This is an important detail that needs to be clarified.

Response: Thanks for your helpful suggestion, which is essential for us to improve the quality of our paper. For the setting of HS method parameters, we have added relevant references in the text, as follows:

Geem, Z.W.; Kim, J.H.; Loganathan, G.V. A new heuristic optimization algorithm: harmony search. Simulation. 2001, 76 (2), 60–68.

Kim J.H.; Geem Z.W.; Kim E.S. Parameter estimation of the nonlinear muskingum model using harmony search. J. Am. Water Resour. Assoc. 2001, 37(5), 1131-1138.

Omran Mahamed G.H.; Mahdavi M. Global-best harmony search. Applied Mathematics and Computation. 2008, 198(2), 643-656.

For the setting of KELM method parameters, we have the convergence curve of RSME as the objective function when the number of hidden layers is 20, which provides calculation support for setting the structure of the KELM model is in the form of "6-20-1", and we have drawn Figure 6.

Thank you for the things you did for our manuscript again! We are looking forwarding to your reply!

Best regards,

Dr. Wang
